# Carbapenem-Resistant Organisms Isolated in Surgical Site Infections in Benin: A Public Health Problem

**DOI:** 10.3390/tropicalmed7080200

**Published:** 2022-08-21

**Authors:** Carine Laurence Yehouenou, Reza Soleimani, Arsène A. Kpangon, Anne Simon, Francis M. Dossou, Olivia Dalleur

**Affiliations:** 1Clinical Pharmacy Research Group (CLIP), Louvain Drug Research Institute (LDRI), Université catholique de Louvain, Avenue Emmanuel Mounier 73, 1200 Brussels, Belgium; 2Laboratoire de Référence des Mycobactéries (LRM), Cotonou BP 817, Benin; 3Faculté des Sciences de la Santé (FSS), Université d’Abomey Calavi (UAC), Cotonou BP 526, Benin; 4Department of Laboratory Medicine, CHU UCL Namur, 5530 Yvoir, Belgium; 5Ecole nationale des Techniciens Supérieurs en Santé Publique et Surveillance Epidémiologique, Université de Parakou, Parakou BP 123, Benin; 6Centres hospitaliers Jolimont, Prévention et Contrôle des infections, Groupe Jolimont asbl, Rue Ferrer, 7100 Haine-Saint-Paul, Belgium; 7Department of Surgery and Surgical Specialties, Faculty of Health Sciences, Campus Universitaire, Champs de Foire, Cotonou BP 188, Benin; 8Pharmacy, Clinique universitaire Saint-Luc, Université catholique de Louvain, UCLouvain, Avenue Hippocrate 10, 1200 Brussels, Belgium

**Keywords:** carbapenem-resistant organisms, OXA-48, NDM, VIM, surgical site infections, Benin

## Abstract

An alarming worldwide increase in antimicrobial resistance is complicating the management of surgical site infections (SSIs), especially in low-middle income countries. The main objective of this study was to describe the pattern of carbapenem-resistant bacteria in hospitalized patients and to highlight the challenge of their detection in Benin. We collected pus samples from patients suspected to have SSIs in hospitals. After bacterial identification by MALDI-TOF mass spectrometry, antimicrobial susceptibility was performed according to the Kirby–Bauer method. Carbapenem-resistant strains were characterized using, successively, the Modified Hodge Test (MHT), the RESIST-5 O.K.N.V.I: a multiplex lateral flow and finally the polymerase chain reaction. Six isolates were resistant to three tested carbapenems and almost all antibiotics we tested but remained susceptible to amikacin. Four (66.7%) of them harbored some ESBL genes (bla_CTX-M-1_ and bla_TEM-1_). The MHT was positive for *Carbapenems* but not for *Pseudomonas aeruginosa* and *Acinetobacter baumannii.* As surgical antimicrobial prophylaxis, five of the six patients received ceftriaxone. The following carbapenems genes were identified: bla _OXA-48_(33.3%, *n* = 2), bla_NDM_ (33.3%, *n* = 2) and bla_VIM_ (33.3%, *n* = 2). These findings indicate a need for local and national antimicrobial resistance surveillance and the strengthening of antimicrobial stewardship programs in the country.

## 1. Introduction

Surgical site infections (SSIs) are one of the most frequent types of healthcare-associated infections (HAIs). They can significantly increase both clinical and economic burdens [1], which could be explained by the direct costs of prolonged hospitalization and the increasing rates of antimicrobial resistance [2]. Of particular concern to surgeons are extended-spectrum beta-lactamases (ESBL) and carbapenem resistance in Gram-negative organisms [3]. This is especially the case in resource-limited countries, where constrained health infrastructures and limited antibiotic options mean that their spread would have devasting consequences. Due to the lack of new alternative treatment options, carbapenem resistance in Gram-negative organisms—especially *Enterobacterales, Pseudomonas aeruginosa* and *Acinetobacter baumannii*—has thus become a matter of concern worldwide [4].

Resistance to carbapenems can be caused by the production of carbapenemases (class A: *Klebsiella pneumoniae* carbapenems (KPC); class B: metallo-beta-lactamases (NDM, VIM IMP-types); and Ambler class D: oxacillinases (OXA-48)). Their spread is mainly due to plasmids acquisition or other mobile elements, to permeability alterations caused by the loss of porins, and to an overexpressed efflux system [5]. The most critical mechanism underlying resistance to carbapenems is the production of carbapenemases: an enzyme that can hydrolyze almost all beta-lactam antibiotics [6]. Another concern is the broad-spectrum resistance of *P. aeruginosa*, which is mainly due to a combination of four factors: (1) low outer-membrane permeability; (2) the presence of the inducible AmpC chromosomal beta-lactamase; (3) the synergistic action of several multidrug efflux systems; and (4) the presence of transferable resistance determinants, particularly carbapenem-hydrolyzing enzymes [7,8,9].

The accurate and timely detection of carbapenem-resistant *Enterobacterales (CRE),* especially carbapenemases-producing *Enterobacterales (CPE)*, is important for the prevention and treatment of such infections [10]. However, despite the therapeutic implications of knowing whether an organism produces carbapenemases [11], most clinical microbiology laboratories still do not yet characterize the mechanism behind carbapenem resistance. Although antimicrobial susceptibility testing (AST) results alone are often enough to support the selection of appropriate antibiotic therapy, it seems necessary to identify the carbapenemases mechanism, particularly when access to AST for newer agents is limited. This is the case with antibiotics such as ceftazidime-avibactam or meropenem-vaborbactam, which are active against some carbapenemases (such as *Klebsiella pneumoniae* carbapenems (KPCs)), but not against others (such as metallo-beta-lactamases {MBLs}, including the New Delhi metallo-beta-lactamases {NDMs}) [12].

Although the spread of carbapenem-producing *Enterobacterales* is now being reported at alarming levels worldwide, very little data are available for Africa, particularly West Africa [11]. In Benin, for example, the lack of a national antimicrobial surveillance system and the resulting shortage of quality data make it difficult to evaluate carbapenem resistance [13]. Given these considerations and the lack of information globally, the purpose of this study was to determine the patterns of carbapenem-non-susceptible organisms isolated in surgical site infections in Benin and to summarize the challenges of their detection.

## 2. Materials and Methods

### 2.1. Ethics Statement

This study was part of a larger project entitled “Multidisciplinary Strategy for Prevention and Infection Control” for which ethical approval was obtained from the Faculty of Health Sciences (Cotonou, Benin) under reference number: 012-19/UAC/FSS/CER-SS. Written informed consent was sought from each participant before enrolment in the study.

### 2.2. Study Design, Bacterial Isolation and Culture

A descriptive cross-sectional study was designed and carried out to determine the epidemiology of surgical site infections in Benin. This study is part of a larger project (Multidisciplinary Strategy for Prevention and Infection Control: MUSTPIC) that was conducted from April 2018 to January 2020 to explore the etiological bacteria involved in surgical site infections in Benin. For one year in this period (January 2019 to January 2020), we included two wards (maternity and gastrointestinal) at six public hospitals. These wards were chosen for two reasons: 1) cesarean sections are one of the most common surgical procedures in Benin, and 2) all hospitals have a gastrointestinal ward. During the study period, all patients ≥ 18 years of age and presenting SSIs according to the Centers for Diseases Control and Prevention (CDC) criteria [14] were offered inclusion in the study. The eligible patients who fulfilled the inclusion criteria were identified by a trained nurse’s team and were subject to wound sampling according to the standard operating procedures (SOPs). Sociodemographic and clinical data were collected by using a standardized questionnaire. A total of 304 wound swabs were collected from 174 patients with clinical signs of surgical infections. Gram-negative bacteria identification was performed according to the guidelines for microbiological methods of the European Committee of Antimicrobial Susceptibility Testing guidelines [15]. We used various biochemical tests: oxidase and characteristics of the Analytical Profile Index (API 20E, Biomerieux, Lyon) such as the Voges Proskauer (VP) test, indole test and citrate utilization. Cultures were incubated for a total of 48 h (if there was no growth at 24 h) at 37 °C in an aerobic atmosphere, and then examined for microbial growth. All identifications were confirmed in Belgium (Cliniques universitaires Saint-luc) using MALDI-TOF (Matrix-Assisted Laser Desorption Ionization-Time of Flight) mass spectrometry (Brucker Daltonics, Bremen, Germany), employing a threshold of ≥2.0. For this study, only Gram-negative bacteria were concerned. The Gram-positive bacteria will be analyzed and published elsewhere. Figure 1 summarizes the selected CRO amongst carbapenem non-susceptible isolates.

### 2.3. Antimicrobial Susceptibility Testing (AST)

The following antibiotics (Biorad, Marnes-la-Coquette, France) were tested for *Enterobacterales:* ampicillin (10 µg), cefotaxime (30 µg), gentamicin (10 µg), ciprofloxacin (5 µg), trimethoprim + sulfamethoxazole (25 µg), cefoxitin (30 µg), amikacin (30 µg), gentamycin (10 µg), ciprofloxacin (5 µg), ceftriaxone (30 µg), cefotaxime (30 µg), piperacillin (100 µg), tobramycin (10 µg), imipenem (10 µg) and chloramphenicol (30 µg). For non-fermenting bacteria, we also included ceftriaxone (30 µg), ceftazidime (30 µg), piperacillin + tazobactam (110 µg) and ticarcillin + clavulanic acid (85 µg). The results were interpreted according to the latest EUCAST breakpoints (www.eucast.org/clinical_breakpoints (accessed on 25 July 2022)) guidelines. In brief, after 24 h of incubation at 35 °C, the inhibition zones were measured, and the results were analyzed and interpreted. Isolates that were resistant to at least three different antimicrobial classes were considered as multidrug resistant bacteria [16].

### 2.4. Phenotypic Confirmation Test for Carbapenemases Production

Detection was completed using a Modified Hodge Test (MHT): an obsolete test in which an overnight culture suspension of *Escherichia coli* ATCC *(American Type Culture Collection)* 25,922 adjusted to 0.5 McFarland standard was inoculated on an MH agar plate and allowed to dry for 3–5 min. After drying, a 10-µg imipenem disk (Biorad, Marnes-la-Coquette, France) was placed at the center of the plate, and the test strain was streaked in a straight line from the edge of the disk to the periphery of the plate. The plate was incubated aerobically overnight at 35 °C. The isolate was considered as positive for the production of carbapenemases if, within the inhibition zone of the imipenem susceptibility disc, there was a clover-leaf type indentation at the intersection of a tested organism and the *E. coli* 25,922 [17]. CLSI recommended, since 2020, the use of CarbaNP, mCIM and eCIM tests [18].

A second confirmation of carbapenemases production was performed in Belgium by using the RESIST-5 O.K.N.V.I [19]: a multiplex lateral flow assay for the identification of five common domestic carbapenemases (OXA-48, KPC, NDM, VIM and IMP) genes.

### 2.5. Quality Control

Standard operating procedures (SOPs) were strictly followed during all analytical procedures, starting from sample collection, isolation, identification and antibiotic susceptibility testing. All culture media were prepared according to the manufacturers’ instructions and were checked for their sterility and performance. As reference strains for quality control of the antimicrobial susceptibility and biochemical tests, we used two international control bacteria strains: *E. coli* ATCC 25,922 and *S. aureus* ATCC 25923.

### 2.6. Genotyping of ESBL and Carbapenemases-Producing Organisms

A single colony was suspended in 200 µL of distilled water. Before extraction, 10 µL of a McFarland equivalence turbidity standard 3.0 of *A. baumannii* ATCC 19,606 was added to the suspension as an internal extraction control. The samples then boiled at 95 °C for 10 min on a heating block. The 25 µL amplification mixture was then prepared containing 2 µL of DNA extract; 2 × 12.5 µL of master mix multiplex PCR kit (Qiagen Benelux, Antwerp, Belgium); and 200 µM of each primer. The carbapenem-resistant genes (bla_OXA-48_, bla_NDM_, bla_VIM_) were detected using an in-house multiplex PCR set up in a microbiology laboratory (Cliniques universitaires Saint-Luc Brussels, Belgium) according to the procedure published by Bogaerts et al. [20]. Table 1 summarizes different primers and probes used for the multiplex PCR in the laboratory.

## 3. Results

### 3.1. Identification and Antimicrobial Susceptibility Results

Among 304 pus samples, we identified 229 strains. While Gram-positive bacteria represented 21.4% (*n* = 49) of isolates, 78.6% (*n* = 180) were Gram-negative. Altogether, twelve isolates were resistant to at least one tested carbapenem, and six were resistant to the three tested carbapenems (meropenem, ertapenem and imipenem). These isolates were as follows: *Escherichia coli* (*n* = 2), *Pseudomonas aeruginosa* (*n* = 1), *Pseudomonas mendocina* (*n* = 1), *Enterobacter cloacae* (*n* = 1) and *Acinetobacter baumannii* (*n* = 1). Isolates shared a multidrug-resistant phenotype. They were resistant to carbapenems and almost all beta-lactam antibiotics including broad-spectrum cephalosporins, amoxicillin/clavulanic acid and fluoroquinolones, but only susceptible to amikacin. Figure 1 shows the flowchart of the study.

### 3.2. Results of Modified Hodge Test (MHT) and RESIST-5 O.K.N.V.I

The MHT was positive for *Enterobacterales* (*E. coli* and *E. cloacae*) only. The test was not performed for non-fermentative bacteria.

The multiplex lateral flow immunochromatographic test showed positive tests for OXA-48, NDM and VIM carbapenems.

### 3.3. Molecular Detection of ESBL and Carbapenemases Genes

Amongst 12 isolates that presented at least resistance to one carbapenem, six were phenotypically confirmed as ESBL and carbapenemases producers’ organisms and carried at least one carbapenemase gene. Four isolates (2 strains of *Escherichia coli*, *Enterobacter cloacae* and *Pseudomonas aeruginosa*) co-harbored ESBL and carbapenemases genes. The two *Enterobacterales* (*E. coli* and *E. cloacae*) carried three ESBL genes (bla_TEM,_ bla_OXA_ and bla_CTX-M_). The non-fermentative bacteria (*Pseudomonas* and *Acinetobacter*) encoded metallo-beta-lactamases and, respectively, VIM and NDM. Except for two isolates, the four others showed at least two beta-lactamases genes. The following tables successively showed the socio-demographic characteristics of included patients (Table 2) and the molecular results of carbapenem resistant strains and their antimicrobial susceptibility (Table 3).

## 4. Discussion

Due to treatment failure and the limitations of alternative therapeutic options, the emergence and dissemination of carbapenem-resistant Gram-negative bacteria remain a challenge for clinicians/healthcare infrastructures. In the present study, we determined the patterns of carbapenem-resistant isolates recovered from pus samples in Benin and summarized the challenges of detecting them in the laboratory. All of the six isolates were resistant to multiple classes of the antibiotics tested, typically to ertapenem, imipenem, meropenem, gentamicin, ciprofloxacin, trimethoprim-sulfamethoxazole and piperacillin-tazobactam. Although Loqman S. et al. reported a similar antibiotic susceptibility profile in Morocco [21], Mathlouthi et al. [22] observed low resistance to imipenem in Tunisian and Libyan hospitals. Such a contradiction outlines the need for a national surveillance system in Africa to study routinely MDR strains. In a study conducted in 2014 with pediatric patients, the authors showed resistance to co-amoxiclav, cefotaxime, cefuroxime, ceftazidime and cefixime and aztreonam, which are in accordance with our results [23]. In 2020, in Alexandria, in a subset of 65 isolates provided from surgical site infection, half of the Gram-negative bacteria, mainly *Klebsiella pneumoniae, Pseudomonas aeruginosa* and *Acinetobacter baumannii*, were resistant to carbapenems [24]. The resulted epidemiological data should help healthcare services as well as clinicians to prevent surgical site infections.

OXA-48 was detected in two strains of *Escherichia coli*. Initially and most frequently, the blaOXA-48 genes were identified in *K. pneumoniae* strains. However, other carbapenems resistance species have recently been recognized as OXA-48 producers [22], including *Enterobacter* spp., *Klebsiella. oxytoca*, *Escherichia. coli* and *Citrobacter. freundii*. OXA-48 and OXA-48-like enzymes are one of the most common CPE enzymes detected worldwide [25]. The family takes its name from the CPE enzyme first identified—OXA-48—and includes several sequence variants that are transmissible via plasmids [26]. OXA-48 carbapenemases are of major concern due to both the detecting issues and to their association with treatment failure [26]. While glycopeptides and daptomycin are still reliable alternatives to methicillin-resistant *staphylococcus aureus*, treatment options are dramatically limited in carbapenemases-producing Gram-negative bacilli. Carbapenemases-producing Gram-negative are particularly resistant to all or nearly all beta-lactams, and often carry other genes that encode resistance mechanisms to fluoroquinolones and/or aminoglycosides [27].

Due to its capacity for colonizing environmental surfaces and spreading within the hospital, carbapenem-resistant *A. baumannii* imposes a high healthcare burden. It is intrinsically resistant to several classes of antibiotic agent, and few treatment options are available [28]. In our subset of studied MDR strains, we identified an NDM-producing *Acinetobater baumannii*, showing the importance of timely identification [28]. In Mulago Hospital, Kampala (Uganda), 2.7% of *A. baumannii* isolates were carbapenem resistant [29]. At Kenyatta National Hospital in Nairobi (Kenya), 85% of *A. baumannii* were MDR [30]. According to the review published in 2020 by Kindu M. et al. VIM carbapenemases were highly prevalent carbapenemase types among *P. aeruginosa* isolates in Africa [31]. As VIM enzymes are one of the most widespread metallo-beta-lactamases that are commonly associated with class 1 integrons or even plasmids, they contribute to the global spread of this resistance mechanism.

To our knowledge, we detected in this study the first case of VIM-producing *P. mendocina* in Benin. Few cases of human infection by *P.mendocina* have been reported since its first description in 1970 by Palleroni et al. after it had been found in water and soil samples collected in Mendoza province in Argentina [32]. More recently, authors in Argentina also reported two cases of VIM-producing *P. mendocina* infections in two hospitalized severely burned patients [33]. As greater understanding would result from studying the distribution of this enzyme in the *pseudomonas* species, further investigations are needed, not only to confirm the dissemination of carbapenemases-producing bacilli in Benin, but also to boost the establishment of an active national surveillance program.

In a context in which the prevalence of carbapenemases is increasing, the early identification of CPE and differentiation between CPE and non-CPE lead not only to improved clinical outcomes, but also to time and cost savings [34]. Rather than using PCR-based methods, which are cost-intensive, time consuming and cannot detect novel unidentified genes, phenotypic methods seem effective and reliable for routine clinical use, particularly in developing countries. Even though the overnight incubation of the Modified Hodge Test (MHT) increased turnaround time, the test was low-cost and needed no special reagent or equipment. However, even though the MHT was recommended by CLSI in 2009, some authors do not recommend it [35]: MHT is not specific for the detection of all carbapenemases enzymes, most significantly with isolates showing weak positive results and AmpC producers [36]. It also has low sensitivity for certain types of carbapenemases (MBL and some OXA types), and lacks specificity, since ESBL and AmpC producers with porin mutations can give false-positive results [37]. Bonnin et al. used the MHT on 19 carbapenemases-producing isolates and found negative results for all the NDM-producing isolates tested, and only weak positive results for the VIM, IMP and OXA-type producers [38]. Hence, all negative MHT should be cautiously interpreted, based on the epidemiological and local data. This test was recommended by CLSI a long time before, and now is an obsolete method. Nevertheless, in our laboratory this test continues to be performed in the routine. Moreover, imipenem was the most resistant of the three tested carbapenems in this study, and this fact could be explained by the overexpression of the efflux pump and loss of OprD porin that are the most common mechanisms of carbapenem resistance in non-fermentative bacteria, especially *P. aeruginosa.* The phenotypic resistance observed may be attributed to the presence of the broad specificity of drug efflux pumps [39].

The aim of surgical antimicrobial prophylaxis (SAP) is to prevent SSIs by administering an antibiotic that targets the microbes most likely to contaminate the surgical site [27]. It should be noted that employing the antibiotic with the narrowest possible spectrum for the shortest possible period, while achieving adequate and timeous tissue levels for the duration of the surgery, reduces SSIs and adverse effects as well as microbial resistance [40,41]. We observed that the most prescribed antibiotics for prophylaxis were third-generation cephalosporins (Table 2), which were not required for prophylaxis [42] as they have less activity against staphylococcal infections, have high rates of resistance development and impose a higher financial burden on patients than cefazolin [43]. In addition, the emergence of extended-spectrum beta-lactamase-producing microorganisms is due to the overuse of third-generation cephalosporins [44]. Although cefazolin is not available in Benin, the appropriate use of SAP is a proven strategy for reducing surgical site infections, and consequently antimicrobial resistance. Our findings also highlight the importance of implementing the antimicrobial stewardship program, of continued medical education and of making the recommended antibiotics available.

The high rate of antimicrobial resistance in developing countries is related, among other things, to limited antibiotic options: a problem that is compounded by the fact that few antibiotic alternatives are currently in the pipeline. Partly because development of new antimicrobial drugs is not financially attractive [45], the development of AMR is exacerbated by the poor availability, affordability, accessibility, and quality of antibiotic agents. In addition, self-treatment with antibiotic agents is common in low-middle income countries, where antibiotics continue to be dispensed without a doctor’s prescription [46], even though the health authorities do not officially allow their sale if they have not been prescribed by a doctor. The factors associated with AMR also include a lack of data on local antibiotic agents, and fear of treatment failure. In Benin, policies to regulate antibiotic use are insufficient and poorly enforced. As the unregulated use of antibiotic agents in LMICs makes them easily available over the counter, their improper uses contribute to the risk of AMR in the country [47]. Moreover, there is a lack of national guidelines for the practice of SAP in hospitals.

This study highlights once again the importance of early detection of CRO. The Modified Hodge Test is obsolete and should be replaced by recommended tests in our laboratory [18]. This should improve the treatment of patients. The practice of SAP is non-optimal because of the unavailability of national guidelines in surgeries departments, and the lack of a national program for antimicrobial stewardship [48]. The project establishes the first guidelines for the rational use of antibiotics. This guideline should be promoted in all hospitals.

We understand and acknowledge the limitations of the present study. Firstly, the analysis was performed on a small subset of isolates, which may limit the generalization of the findings. Secondly, further molecular investigation using more robust whole-genome approaches should help lead to a better understanding of the dynamics of transmission of carbapenem-producing organisms.

## 5. Conclusions

In this study, all isolates were resistant to all tested antibiotics, except amikacin. Ceftriaxone was the most used antibiotic in surgery. The carbapenem resistant strains harbored at least two beta-lactamases genes. The rapid dissemination of carbapenem-resistant strains represents a major therapeutic and epidemiological threat that requires regular surveillance studies and the implementation of strict hygiene procedures, especially in Benin hospitals, where adherence to internationally accepted infection control policies seem not optimal. Moreover, the promotion of accurate and reliable phenotypic tests should be established to improve the detection of CRO in our laboratory in Benin.

## Figures and Tables

**Figure 1 tropicalmed-07-00200-f001:**
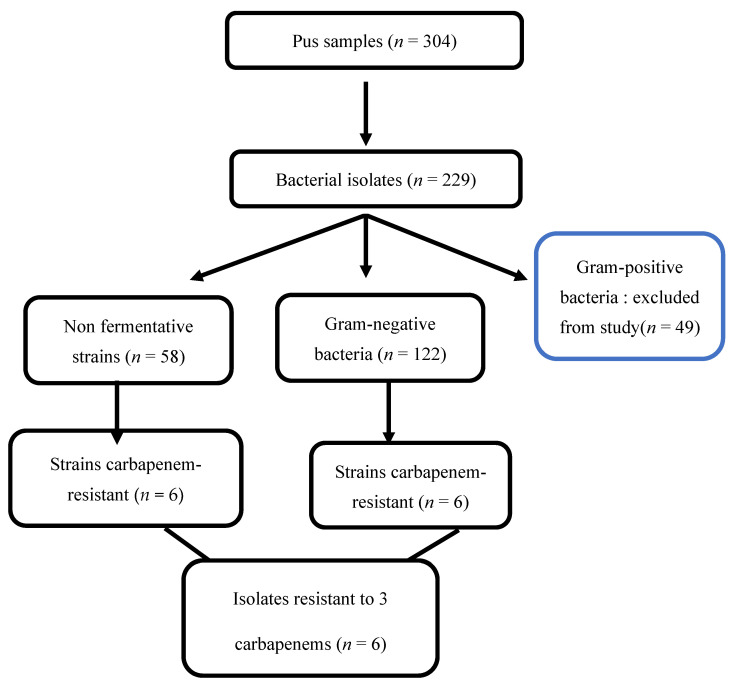
Flowchart of study showing selected carbapenem non-susceptible isolates.

**Table 1 tropicalmed-07-00200-t001:** List of primers and probes used for the in-house multiplex PCR.

Target Gene	Primer Name	Sequence (5′-3′)	Amplicon Size
NDM	NDM-1R	CAT-TAG-CCG-CTG-CAT-TGA-T	596 bp
	NDM-1F-FAM	ACT-TGG-CCT-TGC-TGT-CCT-T	
VIM	VIM437R	ATT-CAG-CCA-GAT-CGG-CAT-C	435 bp
	VIM437R-FAM	TGT-CCG-TGA-TGG-TGA-TGA-GT	
OXA-48	OXA-48 R	CAT-CCT-TAA-CCA-CGC-CCA-AAT-C	265 bp
	OXA-48F-FAM	TGT-CCG-TGA-TGG-TGA-TGA-GT	
CTX-M1	CTXM1R	AGC-TTA-TTC-ATC-GCC-ACG-TT	412 bp
	CTXM1F-FAM	AAA-AAT-CAC-TGC-GYC-AGT-TC	
OXA-1	OXA30R	TAA-ACC-CTT-CAA-ACC-ATC-CGT	388 bp
	OXA30F-FAM	TGG-AAC-AGC-AAT-CAT-ACA-CCA	
TEM	TEM500R	CGG-GAG-GGC-TTA-CCA-TCT-GGC	501 bp
	TEM500F-FAM	CAA-CTC-GGT-CGC-CGC-ATA-CAC-TA	

**Table 2 tropicalmed-07-00200-t002:** Socio-demographic characteristics of included patients.

Samples	Age/Gender	Surgical Antimicrobial Prophylaxis (SAP)	Length of Stay (Days)	Procedure
13150	40/F	ceftriaxone	16	peritonitis
12480	34/F	ceftriaxone	28	cesarean
8062	49/M	ceftriaxone	14	peritonitis
6672	28/F	amoxicillin/clavulanic acid	14	cesarean
5823	35/F	ceftriaxone	14	cesarean
3159	31/F	ceftriaxone	10	cesarean

F: female; M: male.

**Table 3 tropicalmed-07-00200-t003:** Antimicrobial susceptibility test and PCR results for six isolates resistant to three tested carbapenems.

	13150	12480	8062	6672	5823	3159
MHT	**+**	**+**	**+**	*NT*	*NT*	*NT*
Beta-lactamases	CTXM_1_,OXA_1_ TEM, OXA_48_	CTXM_1_,OXA_1_ TEM,OXA_48_	CTXM_1_,OXA_1_- TEM,NDM	CTXM_1_,OXA_1_VIM	VIM	NDM
Name of isolates	*E. coli*	*E. coli*	*E. cloacae*	*P. aeruginosa*	*P. mendocina*	*A. baumannii*
Antibiotics						
Ticarcillin	R	R	R	R	R	R
Piperacillin	R	R	R	R	R	R
Ticarcillin/clavulanic acid	R	R	R	R	R	R
Piperacillin/tazobactam	R	R	R	R	R	R
Ceftazidime	R	R	R	R	R	R
Cefotaxime	R	R	R	R	R	R
Cefepime	R	R	R	R	R	R
Imipenem	R	R	R	R	R	R
Ertapenem	R	R	R	R	R	R
Meropenem	R	R	R	R	R	R
Aztreonam	R	R	R	R	R	R
Gentamicin	R	R	R	R	R	R
Tobramycin	R	R	R	R	R	R
Amikacin	S	S	S	S	S	S
Ciprofloxacin	R	R	R	S	R	R
Levofloxacin	R	R	R	S	R	R
Trimethoprim-sulfamethoxazole	R	R	R	R	R	R

Abbreviation: MHT: Modified Hodge test; NT: not tested and +: positive results.

## Data Availability

Data are available on request from the corresponding author.

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
