# Peer review of "Carbapenem-Resistant Organisms Isolated in Surgical Site Infections in Benin: A Public Health Problem"

_tropicalmed, 2022, doi:10.3390/tropicalmed7080200_

Round 1

Reviewer 1 Report

 The purpose of this study was to determine the patterns of carbapenem-non-susceptible Organisms isolated in surgical site infections in Benin and to summarize the challenges of their detection.

- The abstract is long. please follow the journal guidelines.

- The name of the bacteria needs to be in italic.

- The bacteria name should be written completely for the first time in the manuscript and then abbreviated in the rest of the manuscript.

-  You need to provide information at the beginning of the materials and methods on the number of the collected samples and provide a table that includes information on your samples and their distribution.

- You need to add more information in the bacterial isolation section including conditions.

- Provide correlation analysis between phenotypic and genotypic analysis.

- Move the ethical statement section to the beginning of materials and methods.

- You need to add a statistical analysis section in materials and methods and provide stats in the results section.

- Present the antimicrobial resistance data in a table with providing their prevalence.

- The phenotypic and genotypic resistance need to be provided as percentages.

Author Response

Dear Reviewer1, Thank you for comments and suggestions

Reviewer 2 Report

The manuscript describes the carbapenem-resistant Gram-negative bacterial strains isolated from surgical site infections in Benin. Following major concerns should be addressed before further processing the manuscript.

  1. The manuscript has several formatting errors, such as double spacing between words that should be checked thoroughly. The manuscript should be checked for Grammar and English editing.
  2. The bacterial names should be italicized throughout. Some bacterial names are written in full after being abbreviated (line 263), and the first letter of the genus is not capitalized in some names (line 272).
  3. Line 34-35: The line "two frequently neglected non-glucose-fermenting Gram-negative bacilli, Pseudomonas aeruginosa and Acinetobacter baumannii." gives the impression that the study includes only these two bacteria. In contrast, the study comprises other Gram-negative bacteria as well. 
  4. The gene names should be written correctly, e.g., blaOXA-48blaNDMblaVIM, etc.
  5. Keywords line 56-57: The Oxa-48 should be OXA-48. Other names like NDM and VIM should also be included in keywords.
  6. Line 137-138: Antibiotic susceptibility was performed following EUCAST guidelines 2015 (Reference 14), while the study was conducted between 2019-2020. EUCAST's latest guidelines 2020 should be followed and cited.
  7.  Line 151-152: "Table 3 presents an overview of antimicrobial susceptibilities." This line represents results and should be deleted from here. 
  8. Line 153: The written method "2.3. Phenotypic confirmation test for carbapenemases production" is a substantial concern. The Modified Hodge Test (MHT) was recommended by CLSI a long time before, and now it is an obsolete method. CLSI 2020 recommended the use of CarbaNP, mCIM and eCIM tests (pages 108-128). The citation mentioned in this methodology Lee et al. 2001 is not a suitable citation even for MHT. MHT and the above citation should be replaced with the recommended method.
  9. Line 185-186: The line "The carbapenemases resistance genes" should be replaced with "The carbapenem-resistant genes" 
  10. Line 200-201: "A total of 304 wound swabs were collected from 174 patients with clinical signs of surgical infections" should be mentioned in the methodology.
  11. Line 202: The study focused on Gram-negative bacteria only. Gram-positive bacteria should be mentioned in the exclusion criteria in the methodology.
  12. Line 203-205: "Altogether 12 isolates were resistant to at least one tested carbapenems". Were these isolates resistant to the same carbapenem drug? It has to be clearly mentioned. Moreover, in the discussion section, please explain why 12 isolates were resistant to one carbapenem and which mechanism/ carbapenemase could lead to single carbapenem resistance.
  13. Line 211: Figure 1 does not show the results of carbapenem resistance to any bacterial isolate and is not suitable to cite in this paragraph. The resistance has been mentioned in Table 3. Figure 1 must be a part of the methodology section.
  14. Line 213-215: The reservations regarding the MHT have already been mentioned above. There are recommended methods that can also be used for non-fermenting bacteria.
  15. Figure 1: needs proper labeling, e.g., writing "304 included" does not make sense. It has to be 304 specimens included etc. Also, full names should be used instead of abbreviations. 
  16. Table 2: Under SAP, C3G reflects third-generation cephalosporin. Why not use the exact names of the third-generation cephalosporin antibiotic? Under the Samples, why a citation no. [19] mentioned with 3159? 
  17. Table 3: The heading of Table 3, "Characteristics of samples," is not suitable to write. The first-row heading is also incorrect and misleading. The first column should also include the heading "Antibiotics" before the start of antibiotic results. The ertapenem results are missing in the table. Moreover, none of the bacterial isolates mentioned in Table 3 showed resistance to only one carbapenem. Sulfamethoxazole is mentioned alone in the table while it was used as trimethoprim-sulfamethoxazole. 
  18. Lines 304-306: the authors have cited that MHT is not recommended while they have used this method for phenotypic characterization. 
  19. Line 321: The words "staphylococcus infections" should be "staphylococcal infections."
  20. Line 334: This abbreviation "LMICs," is not defined in the text before.
  21. Conclusion: The conclusion needs to focus more on the study's findings rather than directions.

Author Response

Dear Reviewer 2, 

Please see the attachement the response point by point to your comments 

Reviewer 3 Report

Reviewer Comments to Author(s): 

1. Please write the bacteria names in italics for example in lines 263, 264, and 266; it must be fixed in lines 69, 70, 73. 

2. Please provide the text's reference for lines 104–109. A minimum of a reference for lines 104–106.

     103    In Benin for ex

     104     ample, the lack of a national antimicrobial surveillance system and the 3

        105     resulting shortage of quality data makes it difficult to evaluate car-

        106     bapenem resistance.

3. In the Subheading 2.5 under heading 2.Materials and methods

2.5. Genotyping of ESBL and---- carbapenemase producing organisms.

Please remove the space between ‘and’ and ‘carbapenemase’.

4. In the Subheading 3.1 under heading 3.Results

3.1. Identification and ---- antimicrobial susceptibility results. 

Please omit the space between ‘and’ and ‘antimicrobial’.

5. There must be a brief description for Table 3. There is no explanation of Table 3 throughout the text. The discussion will facilitate a better comprehension of the table.

6. The paper is not well summarized in the end. Just like the discussion section, add a bit of additional language to this. For instance, what inference was made from the results described above?

Author Response

Dear Reviewer 3, 

please see in attachement the point by point response. 

Round 2

Reviewer 1 Report

the authors addressed all the required concerns, however the manuscript will need to be revised for English

Author Response

Thank you for your comments. The manuscript was checked for English before this RE-submission

Reviewer 2 Report

Please see the attached file for comments.
